# The Role of Thermokarst Lake Expansion in Altering the Microbial Community and Methane Cycling in Beiluhe Basin on Tibetan Plateau

**DOI:** 10.3390/microorganisms10081620

**Published:** 2022-08-10

**Authors:** Qian Xu, Zhiheng Du, Lei Wang, Kai Xue, Zhiqiang Wei, Gaosen Zhang, Keshao Liu, Jiahui Lin, Penglin Lin, Tuo Chen, Cunde Xiao

**Affiliations:** 1State Key Laboratory of Cryospheric Science, Northwest Institute of Eco-Environment and Resources, Chinese Academy of Sciences, Lanzhou 730000, China; 2University of Chinese Academy of Sciences, Beijing 100049, China; 3Zhuhai Branch of State Key Laboratory of Earth Surface Processes and Resource Ecology, Beijing Normal University, Zhuhai 519087, China; 4Key Laboratory of Extreme Environmental Microbial Resources and Engineering, Lanzhou 730000, China; 5Key Laboratory of Desert and Desertification, Northwest Institute of Eco-Environment and Resources, Chinese Academy of Sciences, Lanzhou 730000, China; 6State Key Laboratory of Tibetan Plateau Earth System, Resources and Environment, Institute of Tibetan Plateau Research, Chinese Academy of Sciences, Beijing 100101, China; 7State Key Laboratory of Earth Surface Processes and Resource Ecology, Beijing Normal University, Beijing 100875, China; 8College of Geography and Environmental Science, Northwest Normal University, Lanzhou 730070, China

**Keywords:** thermokarst lake, lake expansion, microbial community, sediment-water, co-occurrence network, Tibetan Plateau

## Abstract

**Highlights:**

The co-occurrence network structure of bacteria was more diverse and complex in sediment than in water, while archaea showed an opposite trend at environment under ice.

Microbial diversity increased in MS and SH points, and microbial community composition and microbial network complexity were also different among four points.

The diversity of functional gene *mcrA* and *pmoA* in water was positively correlated with dissolved methane concentration, and the water dissolved methane concentration was positively correlated with the diversity of functional gene *mcrA* but not with *pmoA* in sediment.

The under-ice environment plays a vital role in the methane cycling of the thermokarst lake.

**Abstract:**

One of the most significant environmental changes across the Tibetan Plateau (TP) is the rapid lake expansion. The expansion of thermokarst lakes affects the global biogeochemical cycles and local climate regulation by rising levels, expanding area, and increasing water volumes. Meanwhile, microbial activity contributes greatly to the biogeochemical cycle of carbon in the thermokarst lakes, including organic matter decomposition, soil formation, and mineralization. However, the impact of lake expansion on distribution patterns of microbial communities and methane cycling, especially those of water and sediment under ice, remain unknown. This hinders our ability to assess the true impact of lake expansion on ecosystem services and our ability to accurately investigate greenhouse gas emissions and consumption in thermokarst lakes. Here, we explored the patterns of microorganisms and methane cycling by investigating sediment and water samples at an oriented direction of expansion occurred from four points under ice of a mature-developed thermokarst lake on TP. In addition, the methane concentration of each water layer was examined. Microbial diversity and network complexity were different in our shallow points (MS, SH) and deep points (CE, SH). There are differences of microbial community composition among four points, resulting in the decreased relative abundances of dominant phyla, such as Firmicutes in sediment, Proteobacteria in water, Thermoplasmatota in sediment and water, and increased relative abundance of Actinobacteriota with MS and SH points. Microbial community composition involved in methane cycling also shifted, such as increases in USCγ, *Methylomonas,* and *Methylobacter*, with higher relative abundance consistent with low dissolved methane concentration in MS and SH points. There was a strong correlation between changes in microbiota characteristics and changes in water and sediment environmental factors. Together, these results show that lake expansion has an important impact on microbial diversity and methane cycling.

## 1. Introduction

Global warming has caused rapid degradation of permafrost, resulting in surface collapse and the formation of basins. With the appearance of snow, rainfall, and meltwater, the thermokarst lakes and ponds are formed [1,2,3]. Latent heat from the water in these basins thawed the subsurface ice, leading to subsequent subsidence, deepening, and expansion of open water [4]. In these newly formed aquatic environments, organic carbon is decomposed by microbes in the permafrost layer causing increased atmospheric emissions of methane (CH_4_) and carbon dioxide (CO_2_) [5] and exacerbating global warming.

In recent decades, the rapid expansion of lake area in the TP has attracted considerable attention [6], and it has become a significant environmental change throughout the TP [7]. A number of studies have documented changes in thermokarst lakes across northern hemisphere permafrost and TP regions [8,9]. The findings showed that the activity of thermokarst lakes had generally grown rapidly between 1969 and 2010 and that lateral lakeshore development frequently occurred preferentially in one direction, generating oriented thermokarst lakes [8,10]. In addition, thermokarst expansion might accelerate the emission of carbon stored in permafrost and a significant amount of methane emitted into the atmosphere [11].

Microbial activity plays a vital role in the biogeochemical cycle of carbon in aquatic systems [12], including organic matter decomposition, soil formation, and mineralization [13,14,15]. In addition to regulating greenhouse gas emissions and climate change (Cavicchioli et al., 2019), microbes have been linked to ecosystem productivity, biodiversity, and resilience [15]. Microorganisms in permafrost and thermokarst lakes are exposed to various environmental stresses, such as low temperature, low water activity, high radiation, and freeze-thaw cycles [16]. The rates of the aerobic and anaerobic metabolism of complex microbial communities in thermokarst lakes vary with the composition of water, bottom sediments, content, and the composition of dissolved and suspended organic matters, minerals, pH, depth, nutrients, moisture, ions, as well as on the limnological and hydrochemical properties of the lake [17,18,19,20,21,22]. In addition, methanotrophic microorganisms in thermokarst lakes oxidized the produced methane under both oxic and anoxic conditions, decreasing the net emissions of greenhouse gases into the atmosphere [2,23]. Thermokarst lakes presently contribute almost a quarter (4.1 ± 2.2 Tg CH_4_ year^−1^) of the annual methane emission from northern lakes [24,25]. Understanding methane emissions from thermokarst lakes is significantly important due to the greater global warming potential [25]. Studying microbial processes and microorganism diversity in these processes is important for investigating greenhouse gas emissions and consumption in thermokarst lakes [26,27].

Lakes act as “air conditioner” to regulate the exchange of water vapor, energy, and radiation with the surrounding environment, the water bodies have an effect on local, regional, and even global climate [28]. Water and sediment are different but highly interrelated habitats in thermokarst lakes [29]. The formation of a thermokarst lake is a sediment and water redistribution process triggered by the thawing of ice-rich permafrost [11,30]. Thermokarst lakes can expand vertically and horizontally and finally create different aquatic habitats for metabolic processes by long-term slumping and collapse and are affected by a warming climate in the long run [11,31,32]. The lake expansion may influence underlying surface condition and meteorological characteristics of the surroundings lands, which would further have an important impact on vegetation development and sediment distribution via changing lake morphology [33,34]. As the thermokarst lake expansion through the degradation of lateral permafrost, the source of the sediment changes, changing from redeposited sediments to increased amounts of sediment from older upland deposits and eroding, followed by a more balanced combination of thermokarst lake and upland sources [35]. Thus, additional studies are required to investigate the impact of lake expansion microbial community and microorganisms involved in the methane cycling of the thermokarst lake.

The study of microorganisms in the extreme environments of the biosphere is a vital task for microbial ecology research [36,37]. Many studies have investigated microbial biodiversity, ecosystem function, global biogeochemical cycle, new metabolic processes, and so forth in the lake center where the sediment developed earlier than that in the lake shore [38,39,40]. Previous studies have reported that microbial communities in thermokarst lakes are similar to those in other freshwater bodies but have a more intense methane cycling [12,41,42,43,44]. A growing body of research has studied the methane cycling in thermokarst lakes of Alaska and Canada and showed acetoclastic, hydrogenotrophic, and methylotrophic pathways of methanogenesis [30,45,46]. However, the processes related to methane cycling in lakes are far from being fully understood [47]. Thermokarst lake studies have been mostly conducted in summer, but microbial communities actively functioning below ice during long winters are also important to the ecosystems in northern areas, which are a substantial component of the global carbon cycle. Ice and snow-cover conditions would influence ecosystem energy balances and biogeochemical processes in aquatic environments in cold regions under climate warming. For instance, the availability of light to phytoplankton is controlled by ice, which leads to rapid oxygen depletion while keeping temperatures higher than freezing in the water column below the ice [44]. Emissions of carbon dioxide and methane during long winter periods can account for a large portion of the annual budget [44]. However, microbial communities and methane cycling microorganisms in subglacial environments remain largely unknown [22].

The TP has the largest permafrost in the mid- and low-latitude regions of the world. The permafrost is approximately 1.06 × 10^6^ km^2^ [48], accounting for 40% of the whole TP area. The permafrost carbon pools of TP are important for quantifying regional and global carbon cycling [49]. About 161,300 thermokarst lakes of various sizes were found on the TP, with a total area of 2825.45 ± 5.75 km^2^ [50]. A previous study elaborated that the CH_4_ emissions on the TP of thermokarst lakes exhibited significant spatiotemporal variations [51]. In this study, in order to gain a more detailed understanding of the microbial process between the shallow and deep zones, microbial communities; microorganisms involved in methane cycling; and methane concentrations under-ice environment of a thermokarst lake on the TP were investigated. The present study aims to: (1) explore patterns of microbes in the sediment and water in terms of the community structure, diversity, and co-occurrence network at an oriented direction of expansion occurred obviously in the thermokarst lake; (2) examine the difference in microbial diversity and community structure among four points of the thermokarst lake; (3) explore the relationship between methane concentration and microbes related to the methane cycling. To achieve these goals, we assessed the abundance of the 16S rRNA gene (for bacteria and archaea), functional gene *mcrA* (for methanogens), and functional gene *pmoA* (for methanotrophs) in the thermokarst lake. In addition, the relationship between microbial community and the physicochemical property was studied. The findings of this study can expand the understanding of microbial communities of thermokarst lake and improve the forecasting models of the carbon cycle.

## 2. Material and Methods

### 2.1. Study Site and Sample Collection

Water and sediment samples were collected on 27 January 2021 from a thermokarst lake of the Beiluhe basin in central TP (92.92° E, 34.82° N, Figure 1). The elevation is between 4418 m and 5320 m, the average annual air temperature is −3.8 °C, and the annual precipitation is about 300 mm [8]. The dominant vegetation types are alpine grasslands and alpine meadows, occupying more than 40% of the area [52]. Numerous thermokarst lakes of various sizes were found throughout the flat areas, and they are surrounded by vegetation [53].

We selected four points (the point SH and MS were defined as lake expansion points, at an oriented direction of expansion predominantly occurring in the ENE–WSW direction [8,54], about 3 m and 20 m away from lake margin, respectively, and the other two points were defined non-expansion points, about 40 (MC), and 60 m (CE) away from the SH, Figure 1) from the shore to the center of a mature-developed thermokarst lake, which formed about 890 years old, with an area of 15,373.6 m^2^, and a measured max depth of 2.8 m in 2020 [51,55]. The lake was proved to have been undergoing expansion in recent years, and it was estimated that the expansion time of the point SH and MS was about 2017 and 2000, respectively, according to satellite data and field measurements [8,56]. The center of the lake is deep (CE approximately 1.5 m, MC approximately 1.2 m), and the shore is shallow (SH approximately 0.8 m, MS approximately 1 m). The depth was measured below the ice in winter, and the measured depths of the lake and shore in summer were 2.5–2.8 m and 1.2–1.5 m, respectively. The ice thickness of the thermokarst lake was 45–78 cm. Sampling was conducted using a mechanical pump, which provided power for the ice auger. At each sampling point, three samples of the corresponding water and sediment were collected from the drilled ice holes. Water samples were collected from the top (T), middle (M), and bottom (B) layers. The water samples were collected in 125 mL sterile plastic bottles and filtered through a Millipore Express polyethersulfone (PES) syringe filter with a 0.22 um pore size (Merck Millpore Ltd, Burlington, MA, USA). Water temperature, pH, dissolved oxygen, and conductivity were measured using a portable multiparameter water quality instrument MultiLine 3630 (WTW GmbH, Weilheim, Germany). The pH meter was calibrated using NIST standards of pH 1.679, 4.006 and 6.865. The accuracy of the conductivity probe was evaluated elsewhere, and it is 1% and 2% for TetraCon 925 (for high conductive) and LR925/01 (for low conductive) respectively. A 100 mL syringe with a three-way valve was used to collect 75 mL water and then 25 mL of high-purity nitrogen (99.9% purity) was injected into the syringe. The syringe was shaken vigorously for 2 min until the gas in the syringe reached an equilibrium between water and headspace gas, then it was left to stand for 30 s. After 30 s, 25 mL headspace gas was extracted to foil bags (to minimize their impact on the measurement of greenhouse gas concentrations and was rinsed three times with high-purity nitrogen before each use). The concentrations of dissolved methane were measured by a gas chromatograph (GC-7890B, Shimadzu, Kyoto, Japan) equipped with a flame ionization detector. The sampling tools were sterilized with 75% alcohol and then air-dried for the next sampling. We used sterilized masks and gloves to prevent contamination throughout the whole sampling period. A total of 48 samples (4 matrices [1 sediment and 3 water] × 3 samples × 4 points) were collected during the whole sampling. All samples were stored at −20 °C for the next analysis.

### 2.2. Physicochemical Analysis

Samples for measurements of various ions (F^−^, Cl^−^, SO_4_^2−^, Na^+^, K^+^, Ca^2+^, Mg^2+^), dissolved organic carbon (DOC) and total nitrogen (TN) were gathered in 125 mL low-density polyethylene (LDPE) bottles after being filtered from the water through 0.45 μm PES syringe filters and stored at −20 °C until laboratory analysis [57]. Samples for dissolved inorganic carbon (DIC) measurements were gathered in 125 mL brown, gastight, glass bottles (precleaned using ultrapure water). To prevent biological degradation and photodegradation, DIC samples were poisoned with a 0.2% saturated HgCl_2_ solution and stored at 4 °C in the dark [51]. DIC, DOC, and TN were analyzed with a SHIMADZU TOC-VCPH analyzer (Shimadzu Corp, Kyoto, Japan). The pH and conductivity of sediment were determined using a soil parameter meter (STEPS COMBI5000, Nuremberg, Germany) with a sediment/water (1:5) suspension. The sediment moisture content was gravimetrically measured by drying the 5 g sediment samples with an oven to a constant weight at 105 °C. Approximately 50 g homogenized sediment was freeze-dried with a Labconco FreeZone 2.5 A freeze-dried system (Kansas City, MO, USA), and then sieved by 100 mesh sieves to analyze the total organic carbon (TOC). Finally, we analyzed the TOC of the sediment with the SHIMADZU TOC-VCPH analyzer (Shimadzu Corp., Kyoto, Japan).

### 2.3. DNA Extraction and Sequencing Analysis

The total DNA was extracted from the filtered water and sediment samples using the Power Soil DNA Kit (Qiagen) following the manufacturer’s instructions. Firstly, the extracted DNA was examined on a 1% agarose gel, and a NanoDrop 2000 UV-vis spectrophotometer (Thermo Scientific, Wilmington, NC, USA) was used to determine DNA concentration and purity. Bacterial and archaeal 16S rRNA genes were amplified by using universal primers 338F/806R and 524F10extF/Arch958RmodR, respectively [58], and functional genes (*mcrA* and *pmoA*) were amplified by the primers MLfF/MLrR and A189F/mb661R, respectively (Appendix A) as follows [59]: initial denaturation at 95 °C for 3 min, followed by 27 cycles of denaturing at 95 °C for 30 s, annealing at 55 °C for 30 s and extension at 72 °C for 45 s, and single extension at 72 °C for 10 min, and end at 4 °C. The PCR mixtures contain 5 × TransStart FastPfu buffer 4 μL, 2.5 mM dNTPs 2 μL, forward primer (5 μM) 0.8 μL, reverse primer (5 μM) 0.8 μL, TransStart FastPfu DNA Polymerase 0.4 μL, template DNA 10 ng, and finally ddH_2_O up to 20 μL. PCR reactions were performed in triplicate. The PCR product was extracted from 2% agarose gel and purified using the AxyPrep DNA Gel Extraction Kit (Axygen Biosciences, Union City, CA, USA) according to manufacturer’s instructions and quantified using Quantus™ Fluorometer (Promega, Madison, WI, USA). Purified amplicons were pooled in equimolar and paired-end sequenced on an Illumina MiSeq PE300 platform/NovaSeq PE250 platform (Illumina, San Diego, CA, USA) according to the standard protocols by Majorbio Bio-Pharm Technology Co., Ltd. (Shanghai, China).

The 16S rRNA gene sequences were processed using USEARCH v10.0 [60], and the scripts were written by Liu [61]. Metadata are given in Appendix A. The quality of the paired-end Illumina reads was checked by FastQCv.0.11.5 [62] and then processed with USEARCH in the following steps: merging paired reads and renaming sequences, removing barcodes and primers, deleting low-quality reads, and finding non-redundant reads. Afterward, sequences with high similarity (≥97%) were clustered into the same operational taxonomic unit (OTU) using USEARCH. Representative sequences were classified by the SILVA v123 database [63], and then plastid and non-bacteria were removed for bacteria.

### 2.4. Data Analysis

USEARCH v10.0 was used to analyze the alpha diversity and Bray–Curtis distance-based constrained principal coordinate analysis (CPCoA). The alpha diversity boxplot, CPCoA plot, taxonomy barplot, and taxonomy circus plot were visualized with the Vegan R package. Redundancy Analysis (RDA) was used to access the environment–community relationship with Pacman package in R. A relationship network containing all samples was built to show that ecological clusters are comparable between sediment and water. The Psych package was used to calculate pairwise Spearman correlations. Based on pairwise spearman correlation, node-level topological properties were calculated using Gephi (v0.9.2). Moreover, the Gephi (v0.9.2) was used to visualize the co-occurrence networks. Finally, the datasets presented in this study were submitted to the National Center for Biotechnology Information (NCBI) Sequence Read Archive (https://www.ncbi.nlm.nih.gov/, accessed on 1 September 2021), and the accession number is PRJNA807397.

## 3. Results

### 3.1. The Diversity and Community Composition of the Sediment and Water

A total of 4,854,990 raw reads of the 16S rRNA genes for bacteria were obtained from 48 water and sediment samples, and 1,314,384 high-quality sequences were generated after screening (average, 27,383). Based on 97% similarity sequences, 5958 OTUs were obtained. At the same time, 4,005,308 raw reads of archaea yielded 1,201,392 high-quality sequences (average, 25,029), and 1445 OTUs. The rarefaction curves of bacteria for the OTUs showed that the quantity of the observable species increased with sequencing depth, indicating that sampling volumes were adequate and reasonable, and the sediment had a higher richness than water. In addition, CE had the lowest richness in both sediment and water (Appendix A). For bacteria, the Shannon index was similar in the sediment and significantly higher than in the water. The Shannon index was similar in the water among CE, MC, and MS but significantly higher in the SH (Figure 2A, *p* < 0.05). Analysis of bacterial beta diversity with CPCoA (Bray–Curtis distance) indicated that water microbiota produced four diverse clusters: all water sample were segregated along the second coordinate axis, while the CE and SH in water were separated from MC and MS along the first coordinate axis. Sediment samples were separated from water along the first coordinate axis (*p* = 0.001) (Figure 2C). The CPCoA analysis accounted for 65% of the variance. The rarefaction curves of archaea for the OTUs showed that the quantity of the observable species increased with sequencing depth, indicating that the sampling number was sufficient and reasonable. Sediment had a lower richness than water except for the CE (Appendix A). The Shannon index of archaea in water was lower than that in the sediment, but the difference was not significant (Figure 2B). Analysis of archaeal beta diversity using CPCoA (Bray–Curtis distance) indicated that water microbiota formed four different clusters: the CE in water was isolated from others along the second coordinate axis, while the MS and SH in sediment were separated from CE and MC along the first coordinate axis (*p* = 0.001) (Figure 2D). The CPCoA analysis accounted for 48.2% of the variance.

A total of 63 phyla, 184 classes, 433 orders, 699 families, and 1174 genera of bacteria were identified by comparing the high-throughput sequencing results. Bacterial communities were dominated by the bacterial phyla: Proteobacteria, Bacteroidetes, Firmicutes, Actinobacteriota (Figure 3A,B). The relative abundances of Proteobacteria and Bacteroidetes were higher in water (average 60.17% and 28.74%, Figure 3B) than in sediment (average 23.69% and 17.79%, Figure 3A). In contrast, the relative abundance of the Firmicutes was higher in sediment (average 5.07%) than in water (average 1.25%). Actinobacteria was comparable in sediment (average 7.59%, Figure 3A) and in water (average 7.28%, Figure 3B). In addition, Chloroflexi, Desulfobacterota, and Acidobacteriota also occupied a certain proportion in sediment (average 9.70%, 8.97%, and 5.96%, respectively). A total of 11 phyla, 21 classes, 29 orders, 44 families, and 59 genera of archaea were identified from the whole sample. Archaeal communities were dominated by the phyla (Figure 3C,D): Halobacterota, Crenarchaeota, Thermoplasmatota, and Euryarchaeota. The relative abundance of the Halobacterota was higher in water (average 73.87%, Figure 3D) than in sediment (average 44.20%, Figure 3C). Conversely, the relative abundance of the Thermoplasmatota was higher in sediment (average13.84%, Figure 3C) than in water (average 0.92%, Figure 3D). Euryarchaeota was equivalent in water (average 11.59%, Figure 3D) and sediment (average 12.61%, Figure 3C).

At the bacterial phylum level in the sediment (Figure 4A), the main taxa Firmicutes presented significant differences between different points (*p* < 0.05). Some bacterial phyla with a low relative abundance, such as Patescibacteria, also showed significant differences at different points (*p* < 0.05). The relative abundance of Firmicutes was significantly higher in the CE and presented a decrease trend from CE to SH, but the difference between the MC, MS, and SH was not significant. Unlike Firmicutes, Patescibacteria was significantly more abundant in CE and MC, and there were significant differences between the SH and MS (*p* < 0.05). The relative abundances of dominant bacterial phyla Actinobacteriota, Bacteroidetes, and Proteobacteria were significantly different among points in the water (Figure 4B), which was different from the situation in the sediment. The relative abundance of Proteobacteria gradually decreased from the lake center (CE) to shore (SH) (*p* < 0.05). The taxa Campilobacterota and Bacteroidetes were highest in the CE and significantly different from those in other points (Figure 4B). The relative abundance of Actinobacteriota in the SH of water was the highest and significantly different from those in other points (Figure 4B). Some low relative abundances bacterial phyla Patescibacteria and Verrucomicrobia were also highest in SH and presented significant differences with other points (Figure 4C). Dominant phylum Thermoplasmatota and low relative abundance phyla Aenigmarchaeota and Micrarchaeota showed significant abundance differences among points in the sediment (Figure 4D). Thermoplasmatota was also highest in CE and presented a decreased trend from CE to SH. Micrarchaeota was significantly more abundant in CE and showed significant differences with other points. Aenigmarchaeota in our shallow points (MS, SH) and deep points (CE, MC) presented significant differences (Figure 4D). Thermoplasmatota and Micrarchaeota showed similar distribution patterns in water. The relative abundances of them were highest in CE and had significant differences with other points (Figure 4E,F). Euryarchaeota was less abundant in SH, and the relative abundance was only significant different with MS (Figure 4E,F). Iainarchaeota and Asgardarchaeota were significantly more abundant in CE and showed significant differences with MS and SH (Figure 4F). However, there were no significant differences of bacteria and archaea at different water layers.

### 3.2. Relationship between the Sediment Microbial Community and Environmental Factors

We selected the dominant bacterial phylum (Appendix A) and archaeal phylum (Appendix A) in sediment and water, with environmental factors for redundancy analysis (RDA) ranking between them. Axis 1 and axis 2 of the bacterial community RDA ordination double scale explained 73.16% (62.59%) and 20.49% (28.1%) of the variation in sediment and water, respectively. Proteobacteria was significantly correlated with TOC, pH and moisture in sediment (Appendix A). Proteobacteria was also significantly correlated with DIC, DOC, and TN at the lake center in water (Appendix A). Axis 1 and axis 2 of the RDA ordination double scale of the archaeal community explained 72.76% (42.2%) and 15.63% (28.38%) of the variation in sediment and water, respectively. In both MS and SH of sediment, Crenarchaeota was positively correlated with moisture and pH and was negatively correlated with TOC and Cond (Appendix A). We also found that Euryarchaeota was positively correlated with TN and SO_4_^2−^ (Appendix A).

### 3.3. Methane Cycling of the Thermokarst Lake

Several potential methanogenic microbes were discovered in the thermokarst lake (Figure 5A,B). Unclassified *Methanosarcinaceae* and Euryarchaeota, *Methanobacterium*, *Methanothrix*, *Methanosarcina*, and *Methanospirillum* were presented in both sediment and water samples. The relative abundances of *Methanothrix* and *Methanospirillum* were higher in water (average 32.39% and 11.49%) than in sediment (average 5.92% and 5.57%). In contrast, unclassified *Methanosarcinaceae* and unclassified Euryarchaeota were higher in sediment (average 16.6% and 19.82%) than in water (average 3.23% and 14.15%). *Methanobacterium* and *Methanosarcina* were equivalent in sediment (average 20.23% and 3.98%) and water (average 20.22% and 5.67%). Furthermore, the relative abundance of *Methanospirillum* gradually increased in sediment from the thermokarst lake deep (CE) to the shallow area (SH) (*p* < 0.05 Appendix A). The relative abundance of *Methanosarcina* was the highest in SH and showed a significant difference with MC (*p* < 0.05 Appendix A). Unclassified *Methanosarcinaceae* was significantly more abundant in MS than in other points of sediment (*p* < 0.05 Appendix A). *Methanospirillum* was significantly more abundant in the lake center (CE MC) than in the shallow area (MS SH) of water, and it was the highest in the CE (*p* < 0.05 Appendix A). In addition, the relative abundance of the unclassified *Methanosarcinaceae* in water was significantly more abundant in SH than in other points of sediment (*p* < 0.05 Appendix A). *Methanothrix* was significantly less abundant in CE than other points (*p* < 0.05 Appendix A). The methanotrophic community in the thermokarst lake was dominated by unclassified *Methylococcaceae*, USCγ (upland soil clustersγ) [64], *Methylobacter, Methylomonas,* and *Methylococcus* (Figure 5C,D). The relative abundance of *Methylobacter* and USCγ gradually increased from the thermokarst lake center to the shallow area of both water and sediment, and *Methylobacter* was significantly more abundant than other points (*p* < 0.05 Appendix A). The lowest relative abundance of unclassified *Methylococcaceae* was found in SH of both water and sediment (Figure 5C). However, *Methylomonas* and *Methylococcus* were not the dominant methanotrophic in water (Figure 5D).

Methane concentrations varied with layers and was different among points (Appendix A). Methane concentrations significantly increased from the lake shore to the lake center (*p* < 0.05). Methane concentrations in the top layer at all points (except MC) was the highest but was not significantly different from that in other layers. However, we only recorded the oxygen content of three points (MS, MC, CE Appendix A) and oxygen profile in the summer (Appendix A). We found that oxygen content decreased from the water top to bottom, and it should be lower in the winter owing to ice. Different water dissolved methane concentration were positively correlated with the methanogenic alpha diversity (R^2^ = 0.37, *p* = 0.0004 Figure 6B) and methanotrophic alpha diversity (R^2^ = 0.34, *p* = 0.0002 Figure 6D). Water dissolved methane concentrations were also positively correlated with the sediment methanogenic alpha diversity (R^2^ = 0.37, *p* = 0.02 Figure 6A) but not with the sediment methanotrophs. 

### 3.4. Co-Existing Water and Sediment Show Different Ecological Network Patterns

The number of nodes and edges in the co-occurrence networks (Figure 7) differed across sample points, with more nodes and edges in the water network than in the sediment network for archaea. A total of 260 nodes and 1293 edges were identified in the water network, including 95.51% positive links and 4.49% negative links. In contrast, nodes and edges of bacteria were more in sediment than in water. There were 386 nodes and 8212 edges identified in the sediment network, including 58.67% positive links and 41.33% negative links. The bacterial network in sediment was more complex than in water, indicating more powerful and intricate networks of interaction in sediment ecosystems than in water. The archaeal network showed the opposite results (Figure 7A–D). In sediment and water, the node of bacteria was lower in our shallow points than other points, whereas the betweenness and closeness were higher (Figure 7E,F,I,J). For archaea, the node of water and edge of water and sediment were higher in our shallow points than deep points, whereas the betweenness and closeness were lower (Figure 7G,H,K,L). For this study, the degree of microbial (bacteria and archaea) in sediment was significantly higher than in water.

Based on the values of *Zi* and *Pi*, we calculated the possible topological roles of genera in the bacterial network and OTUs in the archaeal network (Appendix A). We divided all nodes into seven subcategories: ultra-peripherals (R1), peripherals (R2), non-hub connectors (R3), non-hub kinless (R4), provincials (R5), connectors (R6), and kinless (R7). No nodes were classified as R5, R6, and R7 hubs. Most nodes of bacteria and archaea in the sediment network were classified as R2, and they were highly correlated in their corresponding modules. Furthermore, many nodes in the water network were classified as R1, especially in the archaeal network. The bacterial genera *Aquiflexum* and *Leptolinea* were classified as main specialists in the sediment. The nodes of *Roseomonas, Ferruginibacter, Porphyrobacter, Devosia,* and more than 40 species in the water were classified as R1 among modules.

## 4. Discussion

The rapid expansion of lakes has become one of the most important environmental changes across the TP [65]. However, the impact of lake expansion on microbial communities and methane cycling have received less attention, and it was still unknown if lake expansion induced changes in microbial network complexity. In this study, we investigated the co-existing bacterial and archaeal communities in sediment and water and the microorganisms involved in methane cycling from the shallow points and the deep points in a thermokarst lake in winter. The diversity of the bacterial and archaeal community was higher in sediment than in water. Proteobacteria, Bacteroidetes, Firmicutes, and Actinobacteriota from the sediment and water samples were identified by 16S rRNA sequencing [43,66]. The dominant phyla Halobacterota, Crenarchaeota, Thermoplasmatota, and Euryarchaeota in this thermokarst lake were also detected in previous studies [67,68]. The sequencing of the16S rRNA gene combined with the *mcrA* and *pmoA* genes indicated the appearance of methanogens and methanotrophs in all samples. This result suggested that methane cycling existed in both water and the sediment of the thermokarst lake. The concentration of methane dissolved in water was positively correlated with the diversity of water methanogenic and methanotrophic alpha diversity, however, methane concentration dissolved in water was positively correlated with methanogens in sediment only, but not with methanotrophs.

### 4.1. Lake Expansion Affects Microbial Diversity and Community Structure of the Thermokarst Lake

Analysis of bacterial diversity based on Shannon diversity and CPCoA (Bray–Curtis distance) suggested the differences in microbial community structure between water and sediment. The alpha diversity indexes showed that bacterial richness was higher in sediment samples than in water samples, as shown in the previous study on other thermokarst lakes [41]. There were also reported sediment had a much lower bacterial diversity than water on TP [29]. The elevation can influence the water bacterial diversity by the intertwined and opposing effects of increased evapotranspiration (ET), mean annual temperature (MAT), and water temperature [69]. Meanwhile, in the winter season, environmental conditions, such as oxygen and temperature, are changed, which will have an impact on the thermokarst lakes in community structure. The big differences between water and sediment communities also indicated that the water communities were not derived from sediment [26]. At the phylum level, the dominant bacterial phyla Proteobacteria, Bacteroidetes, and Actinobacteria were detected in thermokarst lake in previous studies [43,44,70,71]. Proteobacteria and Bacteroidetes were also the representatives of inland water ecosystems [72]. Proteobacteria accounted for a large proportion [73], and the relative abundance of Proteobacteria significantly decreased from our deep points to shallow points (Figure 4B). Some classes of Proteobacteria can live in oligotrophic ecosystems, such as lake sediments [74,75]. The shore expansion of thermokarst lakes can enable nutrient-rich soils, and the nutrient-rich plants that these soils support, to enter lakes so that lake expansion water has enough sunlight and nutrients, which explains why the relative abundance of Proteobacteria in water was lower in the lake expansion area than in the lake center [76,77]. In addition, it was found that DIC increased the primary production by increasing the abundance of primary producers in inland lakes [78]. As lake expansion with thermokarst alters pH, temperature, and moisture and decreases carbon stocks and nitrogen stocks, with nutrient-rich soils and plants entering the lake, the availability of nutrients increases significantly, which will increase gene diversity and expression intensity [76,79,80]. Lake expansion would also enhance the mineralization of organic carbon and affect the community composition and interactions of related microorganisms in lake sediments with different salinity [81].

It was suggested that there was higher archaeal diversity in sediment than in water, but not significantly. In addition, the archaeal diversity of our shallow points (SH and MS) in sediment was higher than MC and CE. Most archaeal phyla were the Halobacterota, Crenarchaeota, Thermoplasmatota, and Euryarchaeota, which were archaeal communities of Arctic permafrost, subarctic lake, wetland soils, and TP lakes [16,47,67,68,82]. The order of Thermoplasmatota can encode novel copper membrane monooxygenases (CuMMOs), which play important roles in the global carbon and nitrogen cycles [83]. At the genus level, *Methanothrix*, *Methanomassiliicoccus*, *Methanobacterium, Methanosarcina, Methanoregula, Methanoregulaceae,* and *Methanolobus* were found in both sediment and water samples, and *Methanobacterium* and *Methanosarcina* were reported in previous studies [16]. Thermoplasmatota in sediment and in water at SH and MS showed significantly different relative abundance with those at CE and MC, suggesting that the lake expansion had an important role in the microbial pattern of thermokarst lake. Moreover, the microbial diversity was associated with some physicochemical parameters. DOC has been reported to play an important role in the microbial community and function [84]. Halobacterota and Crenarchaeota play essential roles in dissimilatory sulfate reduction [85].

### 4.2. Shifts of Methane Cycling in the Thermokarst Lake

The three key functional groups of microorganisms that regulate the fluxes of methane on earth are the aerobic methanotrophic bacteria; the methanogenic archaea, and their close relatives; the anaerobic methanotrophic archaea (ANME), which represent the special lines of descent within the Euryarchaeota able to activate methane; and SRB able to provide an electron sink [86]. Anaerobic oxidation of methane (AOM) can occur with sulfate, nitrogen oxidizes, organic matter, chlorite, or metals as terminal electron acceptors [25]. The appearance of methanogens and aerobic methanotrophs in all samples by combining sequencing of the 16S rRNA gene with the *mcrA* and *pmoA* genes suggested that methane cycling existed in both water and sediment of the thermokarst lake (Figure 5A–D). Unclassified *Methanosarcinaceae* and Euryarchaeota, *Methanobacterium*, *Methanothrix, Methanosarcina*, and *Methanospirillum* were detected in both water and sediment of the thermokarst lake on TP. Several studies reported the *Methanosaetaceae* and *Methanosarcinaceae* families in thermokarst lakes [16,22,30,87]. Some other methanogens, such as *Candidatus* Methanosarcinaceae, *Methanomicrobiales,* and *Methanomassiliicoccaceae,* were found in water, of which *Methanomicrobiales* was found in both sediment and water of a subarctic lake [47]. *Methanobacterium* and *Methanosarcina* were detected in small thaw ponds of the Canadian High Arctic [87]. *Methanosarcinales* use additional energetic substrates, such as acetate and methylated compounds [88]. Unclassified *Methylococcaceae*, USCγ, *Methylobacter, Methylomonas,* and *Methylococcus* were presented in sediment and water. Type I methanotrophs *Methylobacter* is representative of freshwater lakes and is always found in various lakes and wetland and glacier foreland meadow soils of the Tibetan Plateau [89,90]. Due to the absence of standard primers, it is often underestimated [22,91,92]. *Methylobacter* was favored by aerobic environments and cannot oxidize methane in subglacial environments. USCγ were also typical inhabitants of freshwater lakes. Members of the USCγ cluster show high methane uptake ability and be able to remove methane in the atmosphere [10]. The observed difference in the USCγ cluster could be ascribed to the different environmental conditions, such as soil water content. *Methylomonas* could serve as a bio-filter to mitigate CH4 emissions from permafrost [93]. In general, USCγ, *Methylomonas,* and *Methylobacter* was more abundant in SH and MS with low dissolved methane concentration.

A previous study demonstrated that the methane ebullition fluxes in the deep zones were slightly higher than those in the shallow zones [51]. Many studies have also suggested that methane emissions from other aquatic systems were higher at shallow water depths than at deep depths because of a lower potential for oxidation and shorter transport pathways in the shallow depths [94,95,96]. The top layer was below the ice. This may be due to the long-term freeze, which leads to the accumulation of methane on the surface of the water under the ice. Oxygen content may also influence the methane concentration through methanotrophs. The large, dissolved methane concentrations and oxygen content at the surface of the water, especially of the lake center, demonstrated that methane was not completely oxidized in the water column and accumulated at the surface because water ice impeded the interaction between the atmosphere and the lake. In addition, the dissolved methane concentration in water was positively correlated to the water methanogenic and methanotrophic alpha diversity, and the positive correlation between the concentration of water-dissolved methane and methanogenic alpha diversity, but not to the sediment methanotrophic alpha diversity. This further indicated that methanogens make a greater contribution to methane emissions than methanotrophs. As reported in studies of boreal lakes and high arctic ponds, surface water-dissolved methane concentrations were negatively correlated to the relative number of methanotrophic sequences in surface sediment, indicating that methanotrophs play an important role in the regulation of the release of dissolved methane to atmosphere [26,87,97]. The correlations here were against the relative abundance estimated from boreal and arctic lakes and further investigations using quantitative-PCR and methane functional genes will be needed to validate these relationships in different seasons. In our study, we found sulfate concentration was higher in SH than other points (Appendix A). Sulfate-dependent AOM communities, namely anaerobic methanotrophic archaea (ANME) and SRB, used sulfate to oxidize the methane, which was consistent with the high methane concentration in the CE point. Sulfate-dependent AOM were also the main filter of methane in ocean, drained lakes, and permafrost in the Arctic [67,86,98]. The study also showed that sediment methanogens were dominated by the *Methanobacterium*, and the relative abundance of *Methanobacterium* was lower in MS and SH. This result can explain the low methane concentration in our shallow area. Nitrogen oxidizes, chlorite, and metals will have a greater influence on methane cycling so that we should consider these factors in the future.

### 4.3. The Role of Lake Expansion in Microbiome Complexity of Co-Existing Water and Sediment

Microbial network analysis has always been an important tool for studying underlying interactions between microbial food webs [99]. In this study, we used co-occurrence network analysis to investigate the associations between microbial communities of a thermokarst lake on TP. The network topological parameters of node and edge numbers, and betweenness and closeness were used to assess microbial network complexity, with higher node and edge numbers and smaller betweenness and closeness representing greater network complexity [100]. The results showed that thermokarst bacterial network in sediment was more complex than in water, which was different with glacier-fed aquatic systems between water and sediment [57,99]. These features may include the basic building blocks of freshwater microbial networks. Interactions between microorganisms can be reflected by the network topology. For instance, the degree value represents the level of connectivity between genera and OTUs. These results strongly showed that microbial associations were different among four points, and thus influence the complexity of soil microbial community networks. Furthermore, microbial communities with more complex co-occurrence networks are more resistant to outside environmental stresses than those are with simpler networks [101]. However, reverse results compared with this study in that co-occurrence network of bacterial communities was more complex than that in sediment in thermokarst lakes of the Yellow River source area [29]. Our results also demonstrated that network complexity of bacteria reduced and that of archaea increased in our shallow points. These results were consistent to the observation of similar edaphic factors but with distinct effects on bacterial and archaeal communities [102]. *Zi*-*Pi* scatter plots for all nodes in the different points were generated upon the module network. From ecological perspective, peripheral nodes (R1 and R2) represent specialists, while other nodes are generalists [103]. Previous studies also reported *Porphyrobacter* as generalist aquatic bacterial populations was affected by long term environment stress [104].

This study has potential limitations. Sample size in our lake expansion and non-expansion sites, based on comparing four points with one transect, just fulfills the minimum requirement. Our study cannot reach a general conclusion regarding the impacts of the lake expansion on microorganisms in thermokarst lakes with different seasons. Future work, including more study sites from different thermokarst lakes with other seasons, need to be conducted, targeting the microorganisms and methane cycling. 

## 5. Conclusions

Thermokarst lake expansion can accelerate the emission of greenhouse gases in permafrost and may release a significant amount of methane into the atmosphere, However, most studies presently have been carried out in summer. This study synchronously investigated the microorganisms and dissolved methane concentration under ice of a mature-developed thermokarst lake in Beiluhe Basin on TP to study the impact of the lake expansion on microorganisms and methane cycling. The relative abundance of Proteobacteria decreased and Actinobacteriota increased in the water of our shallow points. MS and SH presented higher diversity of archaea in sediment. Importantly, microorganisms have a significant influence on methane emissions in sediment and water by changing decomposition. For example, USCγ, *Methylomonas*, and *Methylobacter* showed higher relative abundance consistent with low dissolved methane concentration in our shallow points. The changes in microbial communities and networks herein emphasized the importance of microbial communities in supporting carbon cycling in thermokarst lake environments.

## Figures and Tables

**Figure 1 microorganisms-10-01620-f001:**
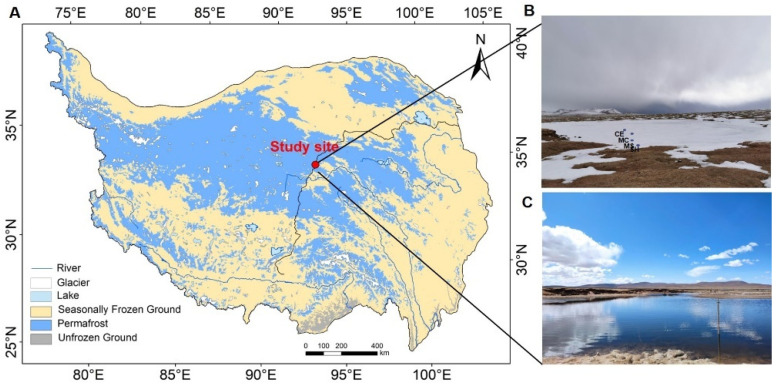
(**A**) Location of the study site in the continuous permafrost of the TP. The frozen-ground map of the TP was plotted with referring to the study of Zou [48]. (**B**) Four sampling points: shallow points (MS, SH), deep points (MC, CE). (**C**) The thermokarst lake in spring.

**Figure 2 microorganisms-10-01620-f002:**
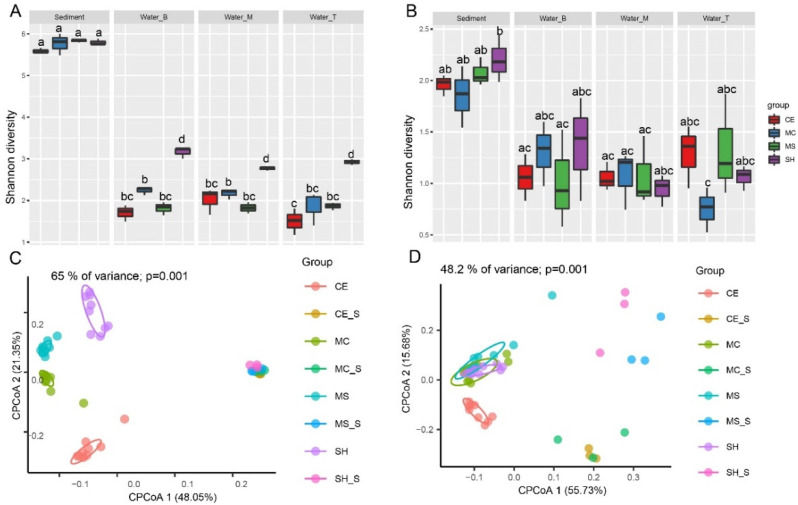
Bacterial and archaeal diversity of water and sediment at different sampling points. (**A**) Bacteria Shannon index in different points; (**B**) Archaea Shannon index in different points; (**C**) Bacterial CPCoA plot using Bray–Curtis dissimilarity based on OTUs in different points (**D**) Archaeal diversity CPCoA plot using Bray–Curtis dissimilarity based on OTUs in different points. Four sampling points: SH (SH in water), SH_S (SH in sediment), MS (MS in water), MS_S (MS in sediment), MC (MC in water), MC_S (MC in sediment), CE (CE in water), and CE_S (CE in sediment).

**Figure 3 microorganisms-10-01620-f003:**
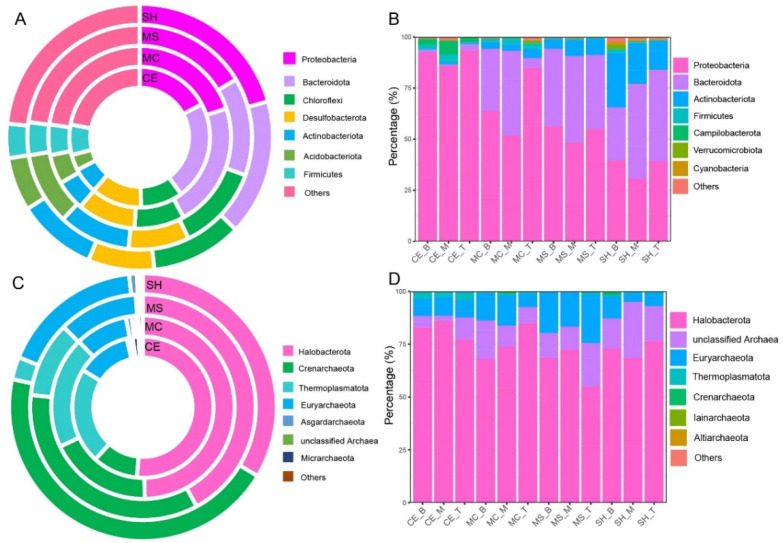
Bacterial and archaeal composition of water and sediment samples from four different sampling points. (**A**) Sediment bacterial composition at the phylum level; (**B**) Water bacterial composition at the phylum level; (**C**) sediment archaeal composition at the phylum level; (**D**) Water archaeal composition at the phylum level. Four sampling points: SH, MS, MC, and CE.

**Figure 4 microorganisms-10-01620-f004:**
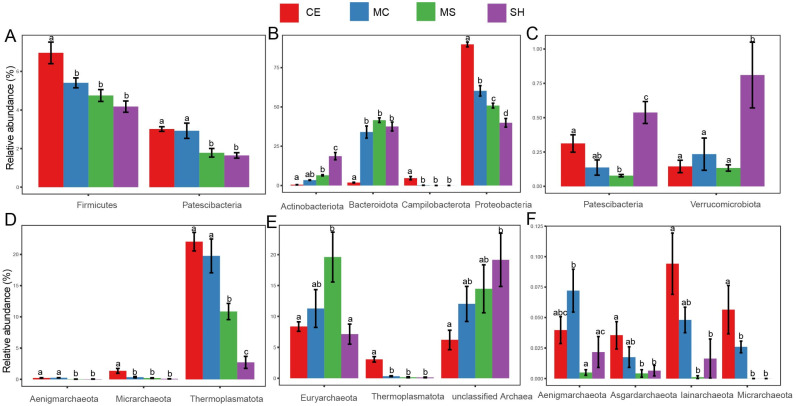
Microorganisms with significant differences among points, (**A**) Bacterial phyla in sediment; (**B**,**C**) Bacterial phyla in water; (**D**) Archaeal phyla in sediment; (**E**,**F**) Archaeal phyla in water. Statistical significance is denoted by differing letters (*p* = 0.05). Columns with the same letters are not significantly different. Different letters meant there was significant difference among points (*p* < 0.05).

**Figure 5 microorganisms-10-01620-f005:**
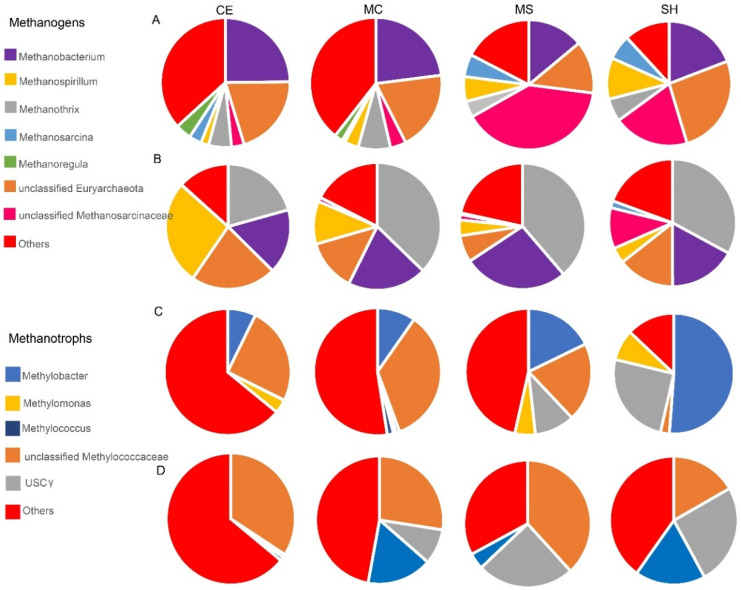
Microorganisms involved in methane cycling. (**A**) Methanogens in sediment at the genus level; (**B**) Methanogens in water at the genus level; (**C**) Methanotrophs in sediment at the genus level; (**D**) Methanotrophs in water at the genus level. Four sampling points: SH, MS, MC, and CE.

**Figure 6 microorganisms-10-01620-f006:**
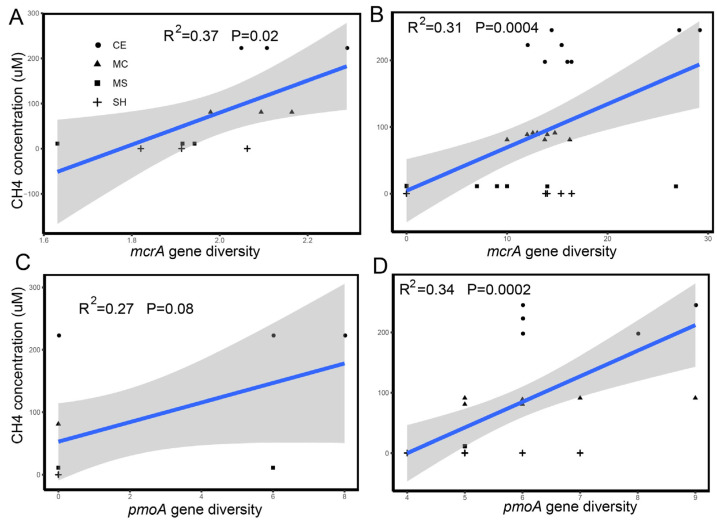
Relationship between the dissolved CH_4_ concentration and alpha diversity of the functional gene *mcrA* and *pmoA*. (**A**,**B**) show the relationship between the dissolved CH_4_ concentration and alpha diversity of functional gene *mcrA* in sediment and water. (**C**,**D**) show the relationship between the dissolved CH_4_ concentration and alpha diversity of functional gene *pmoA* in sediment and water. Significant correlations are shown with a regression line.

**Figure 7 microorganisms-10-01620-f007:**
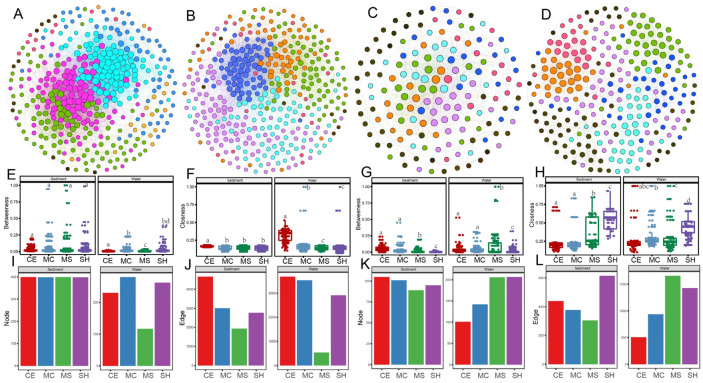
Co-occurrence networks for bacterial and archaeal communities based on pairwise Spearman’s correlations between microbial OTUs water and sediments. The connection edge presents a strong correlation coefficient r > |0.75| and *p* < 0.05. The modules are shown in different colors. (**A**) Co-occurrence networks for bacteria in sediment; (**B**) Co-occurrence networks for bacteria in water; (**C**) co-occurrence networks for archaea in sediment; (**D**) co-occurrence networks for archaea in water. (**E**,**F**) The numbers of betweenness and closeness of bacteria co-occurrence patterns; (**G**,**H**) The numbers of betweenness and closeness of archaea co-occurrence patterns; (**I**,**J**) The numbers of nodes and edges of bacteria co-occurrence patterns; (**K**,**L**) The numbers of nodes and edges of archaea co-occurrence patterns. Statistical significance is denoted by differing letters (*p* = 0.05). Columns with the same letters are not significantly different. Different letters meant there was significant difference among points (*p* < 0.05).

## Data Availability

The datasets presented in this study can be found in online repositories. The names of the repository/repositories and accession number(s) can be found in the article/Appendix A.

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
