# Peer review of "The Role of Thermokarst Lake Expansion in Altering the Microbial Community and Methane Cycling in Beiluhe Basin on Tibetan Plateau"

_microorganisms, 2022, doi:10.3390/microorganisms10081620_

Round 1

Reviewer 1 Report

The manuscript titled: “The role of thermokarst lake expansion in altering the microbial community and methane cycling in Beiluhe Basin on Tibetan Plateau” is very interesting. The authors of the manuscript define the influence of lake expansion on microorganisms in sediments and water in terms of the structure of communities, diversity and coexistence networks of the thermocarst lake. They investigate differences in microbial diversity and community structure at the expansion points of the thermocarst lake and the relationship between methane concentration and microbes related to the methane cycle. Unfortunately, in my opinion, not all chapters of the manuscript have been written correctly and require editing. The Introduction chapter is interesting but too extensive. It does not contain information regarding the title of the manuscript and the research hypothesis. The authors used the correct research methods and received interesting results which are the value of this manuscript. Statistical data analysis is at a sufficient level. In my opinion, the abstract should be slightly edited. The discussion is an important chapter in the manuscript, increasing its substantive value. The research conclusions should be redrafted.

GENERAL COMMENTS

Abstract

In the abstract, please carefully highlight the aim of the research.

Introduction

The Introduction chapter is interesting but too extensive. Information on factors influencing microbiota structural diversity in research ecosystems should be reduced. There is too little information in the introduction about the microbiome, and the authors of the manuscript emphasise that they want to determine the influence of various factors on the microorganisms in the lake and sediments. Please consider shortening the introduction and adding a hypothesis sentence.

Discussion

In my opinion, references to figures in the discussion chapter are redundant.

Conclusion

Conclusions should be redrafted to describe changes in the structure of the microbiome. Conclusions should refer to the title of the manuscript and should correlate with the hypothesis and purpose of the research, which is missing in these conclusions.

Reviewer 2 Report

This study attempts to explore the effect of thermokarst lake expansion on microorganisms and methane cycling on the Tibetan Plateau. Overall, it is an interesting and novel study. Modern (genetic) techniques were employed and bioinformatic analyses were extensive and appropriate. One major problem with the study is that only one lake was sampled. At minimum, this limits the generalizability of the findings; at worst, it results in pseudoreplication. Another issue is some mixing of results and discussion within those two sections. Some English language issues appear, but are fairly minor. I was unable to access the supplementary materials, so am not able to comment on those. Individual comments, both large and small, are detailed below:

Abstract

“explore the effect of lake expansion…” The design of the study (one lake, triplicate samples at each location/depth) allows for statistical comparisons of the four specific locations (distance from shore) and among depths, but does not allow for comparison of shallow vs. deeper areas or “lake-expansion” areas to other areas because only one lake was examined. For instance, any CE vs. SH differences could be due to any number of factors (sediment type, vegetation, etc.) other than depth/newly inundated lake bottom. Short of adding new sampling sites to the study, the authors must take great care with language to avoid pseudoreplication (i.e., stating/implying that they actually have true replicates of lake-expansion locales).

Introduction

l. 99: “content and qualitative composition” please clarify wording

l. 100: add comma after “depth”

l. 103: mean “habitats” here?

l. 111: different than what?

l. 115: remove “are”

l. 134: awkward sentence

l. 138: mean “low-latitude” here?

l. 139: sentence meaning unclear

Methods

l. 172: “at an oriented thermokarst direction” – meaning unclear

l. 206: does 3 copies mean 3 replicates? When were these samples taken?

l. 233: please clarify that same primers for functional genes were/were not used for bacteria and archaea

Results

l. 280: “lowest abundance” – mean lowest species number here?

l. 308: Fig. 4 should not be mentioned in text before Fig. 3

l. 317: not clear what you’re saying here

Fig. 3: why circus graph for sediment, bar for water?

Fig. 6: legend refers to “water layers” but should this say locations (e.g., CE, SH), or is this implying that all the 3 water-depth samples are represented (but not distinguished) here?

l. 430: this is discussion (and there is additional content in sec. 3.3 that belongs in Discussion)

Discussion

l. 468: referring here to the present study or a previous one? Clarify that you also found these groups

l. 475: move “only” to later in sentence

l. 484: define or write out these abbreviations

l. 489: “phylum class” is confusing

l. 495: what’s a “low-trophic environment”? Oligotrophic?

l. 497: need references here

l. 500: meaning unclear

Fig. 8: Figure should be in Results. This work was not described in Methods. Unclear whether same env. variables were measured in sediment and water (appear to be completely distinct). Fig. 8A never referred to in text. Legend has error (A and B assigned twice).

l. 504: Why should nutrient levels be higher? Explain or provide reference.

l. 513: this sentence illustrates the fine line this study treads with respect to pseudoreplication. There is data to show SH differs from CE, etc. (three replicates each) but the study is not able to say diversity is greater in “lake expansion areas” or shallow areas because there is really only a single replicate (the SH site in this one lake) for those comparisons.

Sec. 4.2: several Latin names/terms should be italicized

l. 569: “lake expansion regions”; clearly a problem here given only a single lake expansion site (SH) studied (i.e., pseudoreplication)

l. 576: concentrations of what? Methane? Please clarify

l. 593 (and elsewhere): by “groups” do you mean the 4 sampling areas (distance from shore)? Use of “groups” is confusing.

l. 595: “oxidize”?

sec. 4.3: some results in here, whereas some of the discussion about this work is in Results section

l. 608: how are they different?

l. 624: add “that of”

l. 625: change “that” to “with”

l. 629: “…lake expansion plays a role…” very tenuous statement given N=1

Round 2

Reviewer 2 Report

Some of the problems with the first version of the manuscript have been improved; however, the main point about pseudoreplication is still problematic in my view. As mentioned previously, the reliance on samples from four small areas in one lake (coupled with the correlative nature of the study) make any claims that lake expansion causes the trends observed quite tenuous.

For example, in the Abstract (l. 32) the authors state “Lake expansion increased microbial diversity,” yet this is based on comparing 4 sites (each with 3 replicates collected next to each other) within one lake and assumes any differences are due to lake expansion. It is entirely possible that similar microbial patterns would not have been observed with similar sampling scheme in another lake and/or along another transect in the same lake. (Similar problematic statements are found in l. 33, 36, 39, 49, 159, 637-639, 641, 642, and 644.) Thus, the authors need to be very careful in their language to avoid (implicit) pseudoreplication. My opinion is that all statements such as the above must be re-written in the format “our shallow sites” (or “SH and MS sites” …) instead of “lake expansion areas,” and a short paragraph at the end of the Discussion section is needed to make explicit the limitations of the study (i.e., extrapolating to +/- lake expansion from these four sites on this one transect). Individual comments on other aspects of the (revised) manuscript are detailed below:

Introduction

l. 103: different than what?

Overall, the Introduction is now too long and contains a fair bit of redundancy (esp. paragraphs 3-5).

Methods

l. 185: “at an oriented of expansion occurred obviously” – meaning here is still unclear

l. 184-188: When did the expansion/inundation of the two shallower sites (SH and MS) occur? Can satellite images or other sources be used to estimate the time since expansion? A lot rides on categorizing these two sites as different the two deeper ones, but as far as I can tell we’re relying on a statement that the lakes have expanded recently for that distinction.

l. 222: “irons” to “ions”

l. 275: “Pacman package was used to access the environment-community relationship” Which analyses were employed?

Results

l. 320: move “(Fig.3AB)” to end of sentence

l. 339: “phylum”

l. 350-352: These two (new) sentences are out of place. Further, it is unclear how Proteobacteria relative abundance or DIC relates to primary production (explained later in Discussion)

l. 368-371: these 2 new sentences are also out of place

A separate section is needed to show results of your “environment-community relationship” analyses

Supplemental figures and tables: All could benefit from headings or legends. As is, units are unknown and, at times, it’s not clear what we’re looking at. Table S2 is particularly mystifying.

l. 410-413: this should be in the Discussion

l. 441: Word missing here? “…the degree of microbial (bacteria and archaea) in sediment…

Discussion

l. 479: omit “(Fig.6AC)”

l. 481: move “however” to mid-sentence?

l. 495: add “in community structure”

l. 499: combine two sets of references

l. 503: “Some classes of Proteobacteria can live in oligotrophic ecosystems…” Is this true for other phyla? No relevant references here (the two listed at end of this newly added section do not appear to back up this part of the addition). As written, this argument is weak.

l. 552: should be Candidatus Methanosarcinaceae (italics misplaced)

l. 571: “The” to “A”

l. 598: clarify “…occupied for a smaller part of the thermokarst lake…”

l. 604: “network analysis” or “network modeling”?

l. 618: “postulated” here makes it sound like you are Banerjee et al.

l. 622: Clarify, i.e., “…reverse results compared with this study in that____(ref)”

l. 624-625: I think despite the word “consistent” you’re saying here that prior study showed opposite results(?). Please clarify this.
